# Beam Trajectory Analysis of Vertically Aligned Carbon Nanotube Emitters with a Microchannel Plate

**DOI:** 10.3390/nano12234313

**Published:** 2022-12-05

**Authors:** Bishwa Chandra Adhikari, Bhotkar Ketan, Ju Sung Kim, Sung Tae Yoo, Eun Ha Choi, Kyu Chang Park

**Affiliations:** 1Department of Information Display, Kyung Hee University, Dongdaemun-gu, Seoul 02447, Republic of Korea; 2Department of Electrical and Biological Physics, Plasma Bioscience Research Center (PBRC), Kwangwoon University, Seoul 01897, Republic of Korea

**Keywords:** vertically aligned carbon nanotube, field emission, beam trajectory

## Abstract

Vertically aligned carbon nanotubes (CNTs) are essential to studying high current density, low dispersion, and high brightness. Vertically aligned 14 × 14 CNT emitters are fabricated as an island by sputter coating, photolithography, and the plasma-enhanced chemical vapor deposition process. Scanning electron microscopy is used to analyze the morphology structures with an average height of 40 µm. The field emission microscopy image is captured on the microchannel plate (MCP). The role of the microchannel plate is to determine how the high-density electron beam spot is measured under the variation of voltage and exposure time. The MCP enhances the field emission current near the threshold voltage and protects the CNT from irreversible damage during the vacuum arc. The high-density electron beam spot is measured with an FWHM of 2.71 mm under the variation of the applied voltage and the exposure time, respectively, which corresponds to the real beam spot. This configuration produces the beam trajectory with low dispersion under the proper field emission, which could be applicable to high-resolution multi-beam electron microscopy and high-resolution X-ray imaging technology.

## 1. Introduction

The electron source was one of the crucial components for the development of electron beam microscopy, constructed by Knoll and Ruska in 1932, as its point-to-point resolution was mostly limited by the quality, especially spherical aberration, of the magnetic lenses [1]. In the last 91 years, there have been only four different electron sources used practically in the electron microscope: (i) thermionic emission with tungsten filament, (ii) thermionic emission with lanthanum hexaboride, (iii) Schottky type thermionic emission, and (iv) electric field-induced emission of the tungsten filament. The significant improvement was important for high current density, high resolution, high brightness, and low energy spread [1,2,3]. Considerable attention was paid to the development of the electron source for electron beam microscopy [4,5,6,7,8,9]. In recent years, carbon nanotubes (CNTs) have been considered field emission (FE) materials because they have remarkable properties such as high aspect ratio [10], high electrical conductivity [9,11], high thermal conductivity [12,13], and high mechanical strength [14], which make them extremely attractive as nanoscale reinforcements in high-performance composites. These properties have made CNTs a promising tip material for electron microscopy, e.g., scanning electron microscopy (SEM) [9] and atomic force microscopy [15]. There is great interest in CNT arrays for the successful demonstration of CNT-based field emission displays (FEDs) and field emission lamps. In the fabrication of CNT emitters, oxidation [16], doping [17], and laser irradiation [18] processes have been proposed to improve the field emission of electron beams [19]. The high aspect ratio of CNT emitters play an important role in the field emission electron beam for high stability [11], high brightness [4,20], and low angle of dispersion [21]. The small tip radius of CNTs is most efficient for enhancing the electric field, which reduces the operating voltage of field emission devices [4,9].

Researchers are investigating to improve the FE uniformity and the mechanism of emitter failure of CNT arrays. For example, how to achieve a uniform FE-CNT array; how much the CNT emitter can contribute to the uniform FE performance; when the CNT emitter can be active and destroyed; and how to focus it directly [22,23,24,25]. Some researchers report the forest type of CNT, which is easy to pattern, less time-consuming, and inexpensive, but has many disadvantages, such as not as good field emission performance, the difficulty of focusing compared to the vertically aligned CNTs, and high screening effect [26,27]. The vertically aligned CNTs are proposed to study the uniformity of the field emission with the high efficiency of the electron beam because they have a more uniform geometry [22,24]. These CNT can be fabricated as field emission cathode arrays and used for various purposes such as electron beam microscopy [9], electron beam lithography [28], and X-ray sources [29]. However, variation in CNT size and morphology leads to variation in the local electric field, which may affect the field emission performance of the CNT emitter. The peak point of the CNT emitter is an essential and sensitive part of the uniform field emission.

Conventionally, a phosphor screen is used to observe the FE uniformity of the CNT array. However, this provides insufficient information about the performance of FE, as the homogeneity of the phosphor screen prevents the observation of the actual electron beam trajectory and is affected by the signal noise due to the applied high voltage [24]. When an over-concentrated electric field is applied to the emitter, the tip can destroy itself due to the joule heat effect [30]. To overcome these disadvantages, a microchannel plate (MCP) is used, which provides several advantages. MCP enhances the emission current near the threshold voltage and protects the CNTs from irreversible damage during vacuum arcing [31,32]. MCP operates by avalanche multiplication of secondary electrons generated when an incident electron strikes the MCP’s in-electrode wall. The applied anode voltage of the MCP creates the electric field that accelerates the secondary electrons toward the phosphor screen. The output electron distribution is mapped onto a phosphor screen of the MCP by converging the electron beam signal into photons [33].

In this experiment, 14 × 14 CNT emitters are prepared as one island and considered one beam source to ensure uniform field emission of the highly focused electron beam. The uniform FEM image is captured while the applied voltage and exposure time are controlled. The electron beam trajectory is controlled by the symmetry of the electron beam spot. On the MCP’s phosphor screen, the high-density beam spot increases with the applied voltage but remains constant after 900 V with an FWHM of 2.71 mm. The purpose of this article is to describe how the uniform FEM image is captured by the many CNT emitters under control of the applied voltage and exposure time, as well as the symmetry of the electron beam trajectory with the beam spot size near the threshold voltage and low dispersion. This electron beam produces a real electron beam spot trajectory near the threshold voltage with low dispersion, which could be applicable for high-resolution multi-electron beam microscopy and high-resolution X-ray imaging technology.

## 2. Materials and Methods

### 2.1. Experimental Design of One-Island Carbon Nanotube

A silicon (Si) wafer with a resistivity of 0.001 Ω.cm was used as a substrate for the fabrication of the CNT emitter. A nickel (Ni) catalyst with a thickness of 20 nm was sputtered [34,35] onto the Si wafer for the CNT catalyst using radio frequency (RF). The Ni catalyst was patterned in the specific region of 14 × 14 CNT emitters with its dot, and pitch size of 3 µm, and 15 µm, respectively, using the conventional photolithography process [36,37]. The CNTs were grown in the specified region using direct current (DC) PE-CVD under the suitable conditions of the C_2_H_2_:NH_3_ (18:200 sccm) gas mixture. The cathode and mesh bias voltage was fixed at −600 V, and 300 V at a pressure level of 1.8 Torr, a temperature of 850 °C, and a time of 120 min, respectively [38]. The one-island CNT emitter contains 14 × 14 emitters as one beam source. The height and dot size of CNT growth were controlled by the photolithography process and the growth conditions on the Si wafer substrate. The height of 40 µm and the spot size of 3 µm of each CNT morphology was observed using the SEM, and the tip size is measured to be 50~100 nm. Figure 1 shows the schematic diagram of the experimental setup of the negative voltage supply and electrode system of the field emission electron beam of CNT with 14 × 14 emitters for the image-capturing process using the MCP (Hamamatsu MCP F6959, Japan). The negative high voltage was in the range of 0 to −2 kV for the CNT emitters of the island. The stainless-steel gate electrode (SUS 304) was fixed at a distance of 250 µm from the cathode, which is shown in Figure 1a. The field emission electron beam started to emit at the peak point of the CNT emitter. The aging process of the CNT was maintained to achieve stable field emission of the electron beam, and the electron emission characteristics and uniformity were significantly improved [19]. The field emission electron beam was measured at the anode using the Keithley instrument 6517A (electrometer/high resistance meter, Tektronix). The MCP was replaced by the anode to measure the current. Figure 1b shows the CNT holder and the SEM image of one island CNT of 14 × 14 emitters with an average height of 40 µm and a peak dot diameter of 50~100 nm. In Figure 1c, the left top, right top, left bottom, and the right bottom show the top view of the module in which the CNT substrate is attached, the high voltage electrode, gate guide, and the SEM image of the mesh electrode (SUS 304) with a diameter of 300 µm attached to the gate guide. Figure 1d shows the FEM image on the phosphor screen of the MCP at 900 V. In this experiment, the vacuum chamber was evacuated by the mechanical and turbomolecular pumps with a base pressure of 1.0 × 10^−8^ Torr.

### 2.2. Microchannel Plate Design

The MCP plate is essentially a plate (disc) made of an electrically insulating material (usually glass) containing a hexagonal array of tiny holes, as shown in Figure 2a. The conventional phosphor screen provides insufficient information about the trajectory of the electron beam and is affected by the effect of signal noise due to the applied high voltage [24]. In addition, the CNT emitter can be damaged by the high electric field due to the joule heat effect [24,30]. MCP has more advantages compared to conventional phosphor screens. The incident electron beam is amplified near the threshold voltage by the generation of secondary electrons and protects the CNTs from irreversible damage during the vacuum arc. Figure 2b shows the schematic diagram of the MCP, which contains three electrodes, namely the MCP-in electrode, the MCP-out electrode, and the phosphor electrode. Furthermore, this MCP was used to study the trajectories of the field emission electron beam with the vertically aligned one-island CNT with 14 × 14 emitters near threshold voltage. After applying negative voltage to the CNT emitter, the field emission electron beam started to emit from the peak point of each CNT under the high electric field and accelerated toward the MCP fixed at a distance of 25 mm from the mesh electrode. The effective diameter of this MCP was 28 mm, and the inner electrode was a mesh electrode and was grounded. The MCP-out electrode was placed at a distance of 1 mm from the MCP-in electrode and was biased with a positive voltage of up to 1.5 kV. Moreover, the phosphor was located at a distance of 1 mm from the MCP-out electrode and was biased with a positive voltage of up to 3 kV to convert the electron beam into a light signal. The phosphor electrode converted the electron signal into a light signal. A Nikon D700 digital single-lens reflex (DSLR) camera was used to capture the emitted light signal throughout the field emission process.

## 3. Results and Discussion

### 3.1. Current–Voltage Characteristics of One-Island Carbon Nanotubes

Many researchers have performed experiments [9,21,22,39,40,41,42] and reported the current–voltage characteristics of the field emission electron beam of the different structures of CNT emitters. The I-V diagram describes the field emission performance, which is influenced by the type of material, arrangement, and surface morphology of the emitters [30]. Carbon nanotubes are considered an ideal material for the fabrication of field emitters due to their high aspect ratio, mechanical strength, and chemical stability [39]. In our previous experiment [9], the emission current and brightness of the CNT emitter were investigated with different tip diameters, geometric field enhancement factors (β_geo_), and the number of samples. Due to the high brightness, low threshold voltage, and high stability of the electron beam, we optimized the group 1 sample from our previous experiment for the different purposes of this study [9]. To understand the characteristics of the field emission electron beam profile from an island beam source, a flat anode was used to measure the I–V curve in the high vacuum chamber of this experiment. Figure 3 shows the I–V curve of the field emission electron beam from vertically aligned single-island CNT emitters as a function of the applied voltage. The threshold voltage of the single-island CNT emitter was measured at 810 V with an emission current of 10 nA. After the threshold voltage, the emission current increased dramatically with the applied voltage. The inset in Figure 3a,b shows the Fowler–Nordheim (F-N) plot and the cone-shaped structure SEM image of the vertically aligned CNT with 14 × 14 emitters, where the dot size, pitch size, and height of each CNT emitter are 3 µm, 15 µm, and 40 µm, respectively, that occupy the total area of 39.2 × 10^3^ µm^2^. The F-N plot of ln(I/V^2^) versus 1/V explains the electron beam’s field emission behavior [9,30]. The relationship between the CNT’s morphology and correspondence field emission can be established [9,21].

### 3.2. Measurement of Field Emission Microscopy Image of CNT Emitters

To obtain quantitative information about the microscopic properties of the CNT emitter, the FEM image was examined using the MCP. The single island containing the 14 × 14 emitter worked as a one-beam source in which the emission current drastically increased under the slowly increasing applied voltage as shown in Figure 3. The center exposed area in the FEM pattern reflects the major electron emission from the protruding point at the end of each emitter tip [24]. From the tip to the trunk of the CNT emitters, the surface was smooth, which contributed to enhancing the electric field at the peak point. The trajectory of the electron beam could not be distinguished from each emitter because of the overlap of the electron beams [43,44,45]. Figure 4 shows the FEM image of the one-island CNT with 14 × 14 emitters without a focusing electrode. When the variable negative voltage was applied to the CNT, the emitted electron beam was focused directly on the MCP plate due to the very small gap distance (250 µm) between the cathode and the gate electrode. The applied voltage increased continuously from 800 V to 940 V in the voltage difference range of 10 V so that the electron beam trajectory increased continuously with increased field emission current, as shown in Figure 4. In Figure 4a, there is no image of the electron beam trajectory because there is no emission current flowing. In Figure 4b–o, the field emission electron beam spot is continuously captured. In Figure 4b, the bright spot of the electron beam appears on the center of the phosphor screen of the MCP at 810 V. The FEM image with a high-density bright spot is visible in the center of the MPC. From Figure 4f, the spot size increases continuously until Figure 4k. From Figure 4k–o, its size remains constant with its full width half maximum (FWHM) of 2.71 mm. The uniform size of the high-density bright spot remains constant from Figure 4k–o, but the size of the green spot increases continuously because the divergence of the electron beam was limited by the electron emission from the vertically aligned CNT emitters [46,47]. The vertically aligned CNT shows a small beam divergence of electron beam trajectory at a distance of 25 mm of the MCP with a high-dense bright spot opening angle of 2.9° [9,48,49,50]. Figure 4a–o confirms that the field emission electron beam from 14 × 14 CNT emitters works as a one-beam source and is focused on a point with a high-density bright spot without a focusing electrode near the threshold voltage. Many experiments are repeated to confirm the size of the high-density bright spot. Figure 5 shows the confirmation of the size of the focused electron beam of the 14 × 14 CNT emitters under the same conditions as in Figure 4. In Figure 5a–f, the applied voltage is increased from 850 V to 900 V with an increasing range of 10 V to capture the high-dense bright spot of the FEM on the MCP. In Figure 5f, the size of the high-dense bright spot is calculated to be 2.71 mm with its FWHM. Hence, the symmetrically distributed FEM image is captured under the variation of the applied voltage on the phosphor screen of the MCP.

### 3.3. Spot Size Trajectory Analysis of Field Emission Electron Beam

The pattern of the high-dense bright spot has a circular spot, and the intensity profile of the beam spot can be well described by a Gaussian distribution, G(x)=G(0)+A.exp{−0.5 (x−x0σ2)2}, where A, $x_{0}$, and σ represent the peak intensity, the maximum point of peak intensity and the standard deviation, respectively. The fullwidth half maximum of the electron beam can be expressed as FWHM = $2\sqrt{2ln2\sigma }$, where σ is the standard deviation [21,51]. Figure 6 explains the analysis of the intensity profile of the high-dense electron beam spot of Figure 4. In Figure 6a, the intensity increases continuously up to 900 V, after which it becomes constant and reaches its maximum intensity at 255 atomic units. As the emission current increases, the intensity profile of the dense bright spot expands. The FWHM is expected to become more compact at higher beam voltage [52]. The FWHM measures the actual size of high-density bright spot of the electron beam. The size of the high-density bright spot increases with the increase in the applied voltage from 800 V to 900 V and then remains constant with a size of FWHM of 2.71 mm, as shown in Figure 6b. In our previous experiment [9], the simulated result is explained in detail and compared with the experimental result, the change in the spot size of the beam path was simulated by varying the distance between the gate electrode and the phosphor screen. Figure 6c shows the variation of the beam divergence when varying the distance between the gate electrode and the phosphor screen according to our previous experiment. The electron beam trajectory follows the fitting parameter W_z_ = w_0_ + aZ^b^ where W_z_ is the beam trajectory, and w_0_, a, and b are fitting parameters with values −0.06270, 0.11089, and 0.75970, respectively. The electron emission and the beam trajectory depend on the structure of the emitter of the nano-tip emitter, the gap distance from the cathode to the mesh electrode, and the gap distance from the mesh electrode to the phosphor anode, respectively. Figure 6d is the schematic diagram for the decrease in the size of the electron beam spot from (i) to (v) with the decrease in the distance between the mesh electrode and the phosphor electrode of MCP, which is clearly explained in Figure 6c. In our experiment, the emitter density is high although the beam is highly focused at a point on the MCP without a focusing electrode. Nonetheless, the proposed MCP approach can generate uniform beam trajectories with the high-density beam spot of the CNT emitters with the best morphology structures under the proper field emission.

### 3.4. Noise Effect of Focus Spot Size

Several processes deteriorate the signal-to-noise ratio (SNR) of electron microscopy images, such as the noise of the primary electron beam, the secondary electron beam in MCP, and the noise of the final detection system [53,54]. In the MCP, the input electron energy increases the generation of secondary electrons compared to the noise [54]. For the measurement of the electron beam spot, exposure time plays an important role in the SNR effect [55,56]. The electron beam spot size is captured in a short exposure time to analyze the spot size of the electron beam from the CNT emitter on the MPC’s phosphor screen. The exposure time is the time in which the light can reach the sensor. The focused electron beam is kept at a fixed position and the beam spot size of FEM is controlled by the exposure time [57,58,59]. Figure 7 shows the reduction of the beam spot by reducing the exposure time at a fixed applied voltage of 900 V. In Figure 7a, the FWHM of the high-density bright spot is calculated to be 2.71 mm for an exposure time of 1s. Moreover, in Figure 7c–e, the FWHM is calculated to be 2.71 mm at an exposure time of 1/3, 1/4, and 1/5 s, respectively, showing the real electron beam trajectory at the center of the MCP. As the exposure time of the electron beam increased, the noise effect in the beam spot size also increased. Figure 8 shows the line profile of the electron beam spot from Figure 7a,b, in which the noise effect is explained by the variation of the exposure time. The FWHM of the real beam spot size is 2.71 mm at 1/5 s. This experiment explains the electron beam trajectory with its spot size by optimizing the applied voltage and exposure time. Hence, the SNR is increasing with the variation of exposure time at the optimized voltage. The FWHM of the high-density beam spot is calculated as 2.71 mm at 900 V under the variation of the exposure time. The real beam spot size is very important to obtain the actual electron beam trajectory.

Figure 9a shows a schematic representation of the electron beam trajectory from Figure 7 where the exposure time is reduced by 1/2, 1/3, 1/4, and 1/5 s respectively. Figure 9b shows a simulation of electron beam trajectory in which opera simulation 3D [60] is used to analyze the high-density bright spot with its current density. The simulation is modelled based on an emitter spot size of 3 µm with a height of 40 µm. In addition, the diameter of the mesh hole is set to 160 µm with a thickness of 100 µm. The distance between the cathode and gate, and gate and anode is set to 150 µm, and 1 mm, respectively. Figure 9c shows the simulation results of the electron beam spot from Figure 9b, where the beam is strongly focused on the center of the phosphor screen (with blue color). The red, yellow, and green colors in Figure 9c represent the scattering of the electron beam from Figure 9b, which can act as a noise effect on the phosphor screen.

## 4. Conclusions

Vertically aligned cone-shaped CNTs with 14 × 14 emitters were prepared as an island in the Si wafer substrate by sputter coating, photolithography, and PE-CVD for the high resolution and low dispersion of the electron beam. This island of CNT emitters was called a single beam source. This beam source was perfectly aligned with the center of the gate electrode (SUS 304) to ensure uniform field emission of the highly focused electron beam. The threshold voltage of the single island CNT emitter was measured at 810 V with an emission current of 10 nA. After the threshold voltage, the emission current increased dramatically with the applied voltage. The uniform FEM image was captured by varying the applied voltage and exposure time to study the symmetrically distributed electron beam trajectory and the beam spot size. The high-density beam spot increased continuously under the variation of the applied voltage and remained constant after 900 V with its FWHM of 2.71 mm on the phosphor screen of the MCP. In addition, the FEM image was captured with variation of the exposure time to investigate the noise effect.

## Figures and Tables

**Figure 1 nanomaterials-12-04313-f001:**
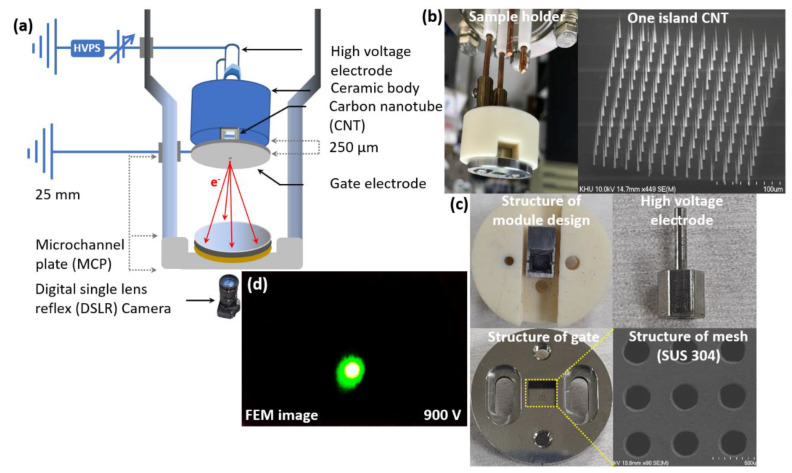
(**a**) Schematic diagram of the experimental setup for the power supply and image capturing process of the field emission electron beam of the vertically aligned carbon nanotubes emitter at the high vacuum chamber; (**b**) The left-hand and right-hand side represent the module setup of the CNT emitter in the high vacuum chamber and the SEM image of cone-shaped vertically aligned CNT containing 14 × 14 emitters, respectively; (**c**) The left (up), right (up), left (down), and right (down) represent the top view of the module in which the CNT substrate is fixed, the high voltage electrode, the structure of the gate module, and the SEM image of the mesh electrode with its diameter of 300 µm attached to the gate electrode, respectively; (**d**) Represents the field emission microscopy (FEM) image at 900 V.

**Figure 2 nanomaterials-12-04313-f002:**
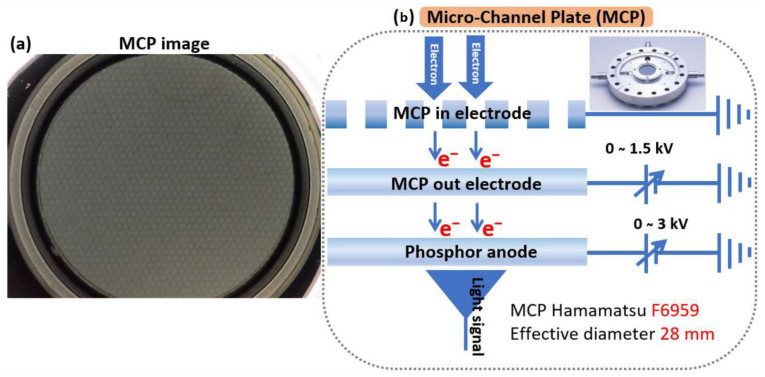
(**a**) The top view of the microchannel plate with the number of holes; (**b**) Schematic diagram of the MCP in which the emitted electron beam is converted to the light signal on the phosphor electrode.

**Figure 3 nanomaterials-12-04313-f003:**
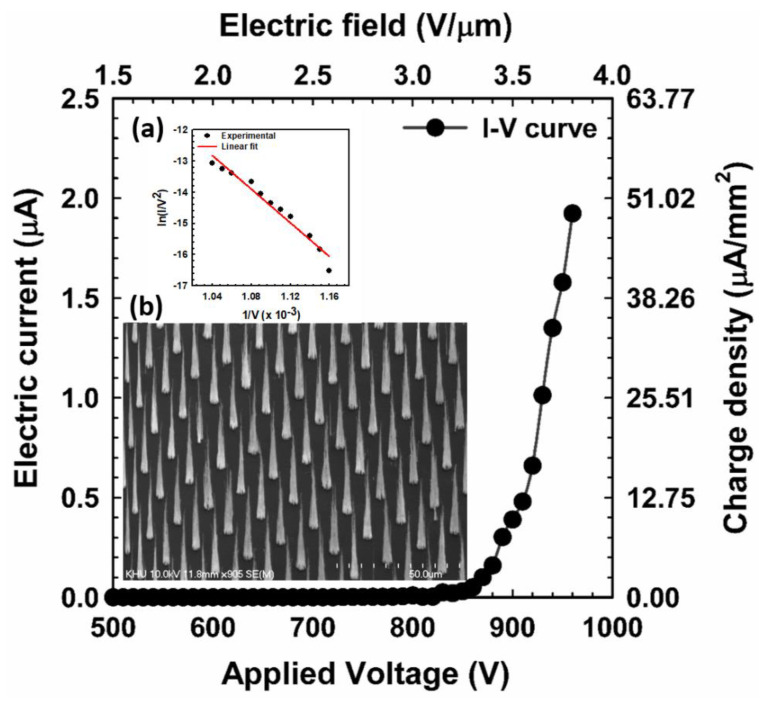
Current–voltage characteristics of the field emission electron beam of vertically aligned one-island CNT containing 14 × 14 emitters. (Inset: (**a**), and (**b**) Represents the F-N plot and SEM image of a cone-shaped CNT structure with 14 × 14 emitters).

**Figure 4 nanomaterials-12-04313-f004:**
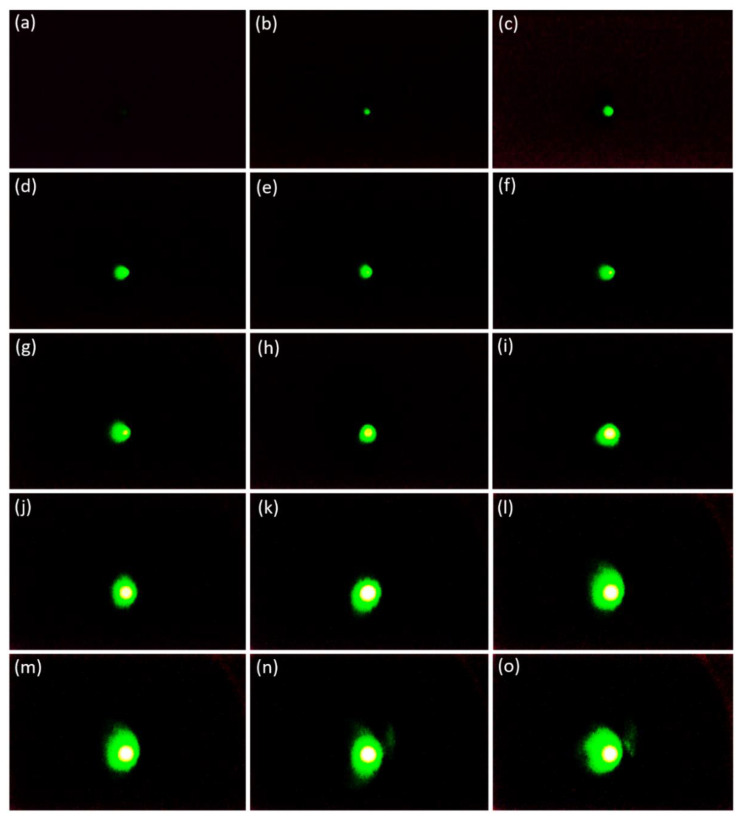
(**a**–**o**) Represents the field emission microscopy image of the one island CNT containing 14 × 14 emitters under the variation of applied voltage 800 V to 940 V with the increasing range of 10 V, respectively.

**Figure 5 nanomaterials-12-04313-f005:**
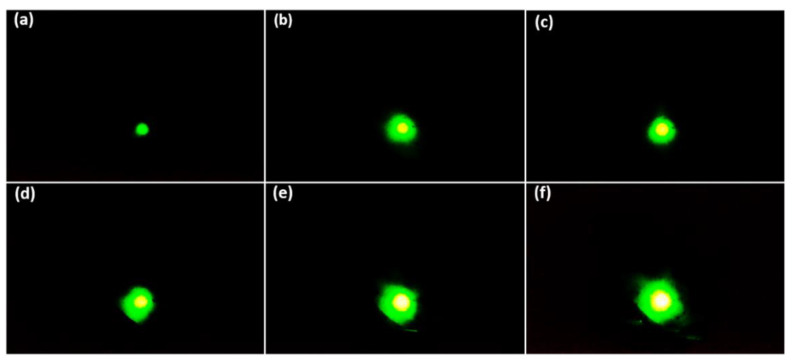
(**a**–**f**) FEM image of the one island CNT containing 14 × 14 emitters under the variation of the applied voltage from 850 V to 900 V under the increasing range of 10 V, respectively.

**Figure 6 nanomaterials-12-04313-f006:**
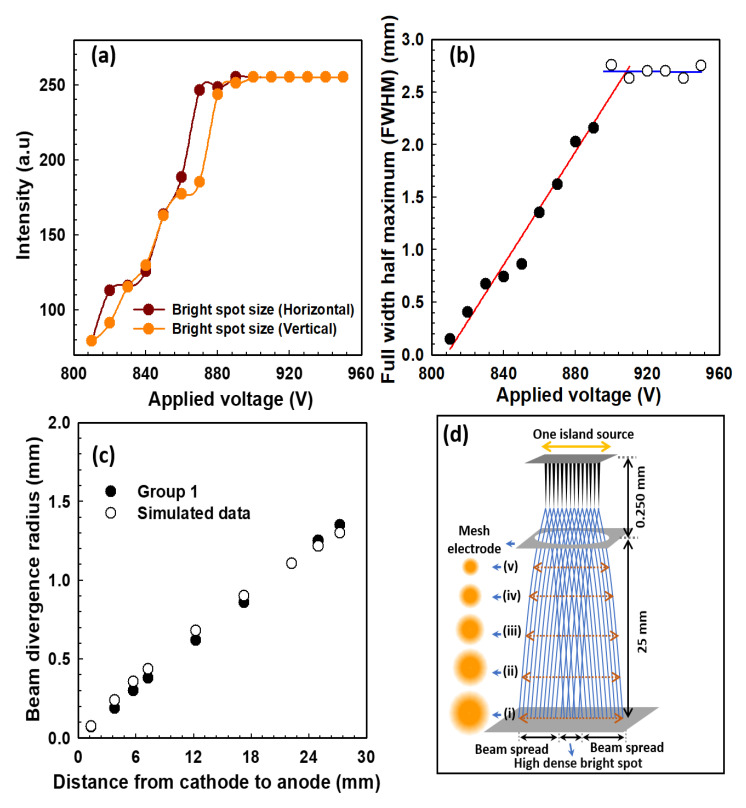
(**a**) Explains the increasing intensity of the high-dense bright spot size under the variation of the applied voltage; (**b**) Shows the full-width half maximum (FWHM) of the high-dense bright spot size under the variation of the applied voltage; (**c**) Shows the beam divergence comparison of simulation results with the experimental results (Group 1), in which beam spot size is increasing as the distance between the gate electrode and the phosphor screen changes; (**d**) The schematic diagram for the reduction of the beam divergence with the variation of the distance from the mesh electrode to the phosphor screen with different beam spot sizes.

**Figure 7 nanomaterials-12-04313-f007:**
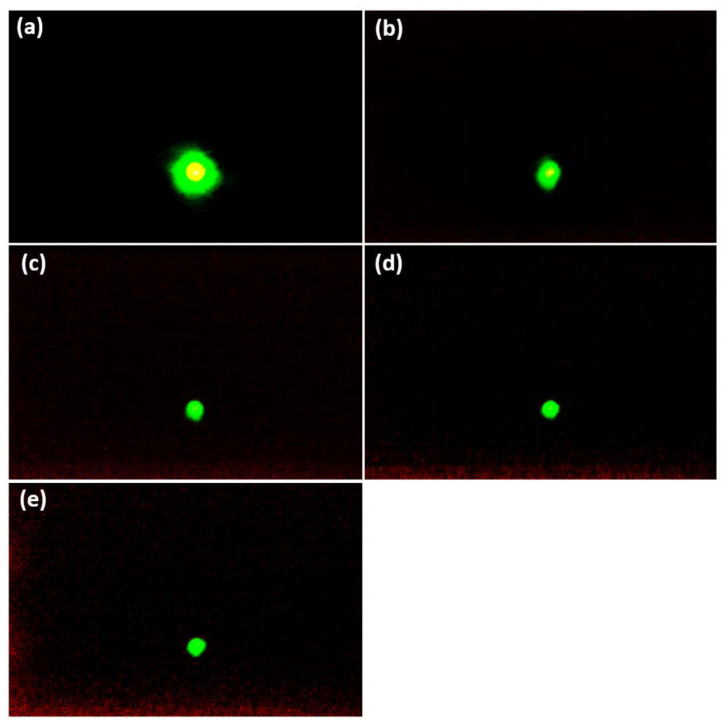
(**a**–**e**) represents the FEM image of the 14 × 14 CNT emitters at 900 V under the variation of the exposure time.

**Figure 8 nanomaterials-12-04313-f008:**
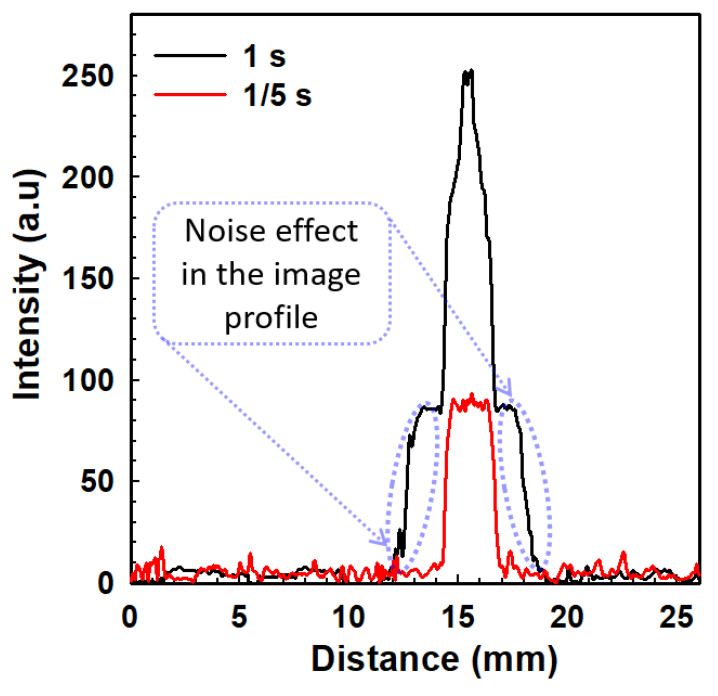
Line profile of the FEM image of Figure 7a,e with the variation of the exposure time under the applied voltage 900 V.

**Figure 9 nanomaterials-12-04313-f009:**
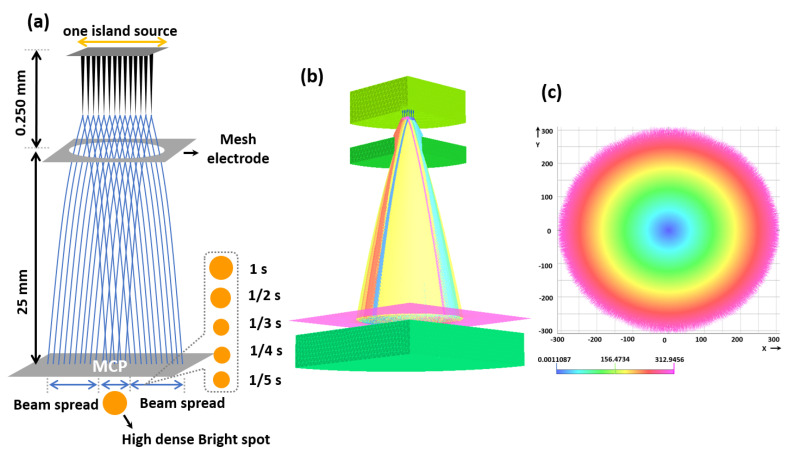
(**a**) Represents the schematic diagram of the electron beam trajectory of Figure 7; (**b**) Represents the simulation results of the electron beam trajectory with its beam spot; (**c**) Represents the electron beam spot on the phosphor screen. The blue color represents the effective electron beam spot.

## Data Availability

The data presented in this study are available on request from the corresponding author.

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
