# Peer review of "Beam Trajectory Analysis of Vertically Aligned Carbon Nanotube Emitters with a Microchannel Plate"

_nanomaterials, 2022, doi:10.3390/nano12234313_

Round 1

Reviewer 1 Report

nanomaterials-2054347-v1

Beam trajectory analysis of vertically aligned carbon nanotube emitters with the microchannel plate

This paper focus into the analysis of carbon nanotube (CNT) emitters that seem to be essential to study the high current, low dispersion, and high brightness. The CNT emitters are fabricated by sputter coating, photolithography, and plasma chemical vapor deposition process. It describes the combination of vertically aligned emitters, the role of the microchannel plate in capturing the field emission microscopy image, and how it is measured the high-density electron beam spot under the variation of the voltage and the exposure time. This configuration produces the beam trajectory under the proper field emission for application in multi-beam electron microscopy and field emission display technology.

The paper is very interesting and there are an important study and description of the elements of the beam source to ensure uniform field emission of the highly focused electron beam. Also, it is detailed how the uniform FEM image is captured under control of the applied voltage and exposure time, and the symmetry of the electron beam trajectory with the beam spot size. They study that high-density beam spot increases with voltage although it remains constant after 900 V with the FWHM of 2.71 mm on the phosphor screen of the MCP.

However, authors fail to show what is the aims of the paper. There are technical and conceptual information together with a historical evolution of the electron sources, but it is difficult to understand how it work and what is more import what the true contribution is. It is not clear if the emitters are obtained in their lab. Authors should work on the introduction and summary to clarify this. Also, through the manuscript they should explain clearly what is done, what is analysed or simulated and what are the source of the materials.

At the end of the introduction, are described experimental conditions and chemical composition of the gas mixture, temperature, etc., with no reference at all. If these conditions have been used in this work should be explained clearer. Perhaps is better to include this description in the experimental section or in results and discussion.

Therefore, I consider that authors need to take care of these points before the paper be considered for publication in nanomaterials.

Minor comments

Please check expressions of 14x14 emitters, 14x 14, and 14 x 14 to write all of them equal.

The sentence on line 102 needs correction.

Capitals starting subsection title. See line 233.

Line 304, reference to the simulation software is needed.

Please, revise spelling

Author Response

Refer the attached file for point-by-point reponse.

Reviewer 2 Report

This paper is related to the beam trajectory analysis of emitters in which the field emission microscopy image has been captured on the microchannel plate and CNTs have been aligned vertically. The topic of the research is interesting and new. The results are original. Some minor points should be considered by the authors:

1- The abstract of the research should be improved. Highlights of the research are unclear. They should be presented separately. In addition, the main contribution of the research should be clearly discussed. 

2- The literature review of the paper should be improved. Some more relevant and recent articles should be considered in this section. 

3- The English language of the text should be double-checked, especially grammatical errors.

4- In the conclusion part, only original results which have been obtained only in this research should be discussed. Some obvious statements in this section are unnecessary and should be removed. 

5- It is better that the time of verb used in the conclusion is past. Because the results have been obtained. 

6-The authors should clarify the main application of the obtained results from the present research in the real work. 

7- Providing a flowchart for the experimental process used in the experimental part is useful and interesting for readers. 

Author Response

Refer the attached file for point-by-point response

Round 2

Reviewer 1 Report

nanomaterials-2054347-v2

Beam trajectory analysis of vertically aligned carbon nanotube emitters with the microchannel plate

This paper focus into the analysis of carbon nanotube (CNT) emitters which are fabricated by sputter coating, photolithography, and plasma chemical vapor deposition process. It describes the combination of vertically aligned emitters, the role of the microchannel plate in capturing the field emission microscopy image, and how it is measured the high-density electron beam spot under the variation of the voltage and the exposure time.

On the revised version, authors have addressed the changes recommended which are explained point by point on their replies. They revised the summary, introduction, some paragraphs through the manuscript, and conclusions. Also, they have included several references in agreement with the comments.

Therefore, after the revision, I recommend the manuscript to be published in nanomaterials.